# Three-Dimensional CBCT Based Evaluation of the Maxillary Sinus by Facial Index

**DOI:** 10.3390/ijerph19095040

**Published:** 2022-04-21

**Authors:** Jeong-Hyun Lee, Jong-Tae Park

**Affiliations:** 1Department of Oral Anatomy, Dental College Dankook Institute for Future Science and Emerging Convergence, Dankook University, Cheonan 330-714, Korea; 911105jh@gmail.com; 2Department of Bio Health Convergency Open Sharing System, Dankook University, Cheonan 330-714, Korea

**Keywords:** maxillary sinus, 3D, facial index, mesoprosopic, leptoprosopic, hyperleptoprosopic

## Abstract

The maxillary sinus growth is initiated 3 months after birth, and it grows lateral and inferior until the pneumatization of the alveolar bone occurs. The facial skeleton has recently been determined as affecting the maxillary sinus, prompting additional studies on changes in the size of the maxillary sinus. This study aimed to determine the size of the maxillary sinus using a 3D program after categorizing South Korean adults according to their facial index (FI) classification. The participants of this study were 60 patients in their 20s, who visited the orthodontic department of Dankook University Dental Hospital (approval no. DUDH IRB 2015-12-022). The CBCT of the patients were extracted and measured as 3D images using Mimics (version 22.0, Materialise, Leuven, Belgium). Upon categorizing the subjects based on their FI classification, they were grouped into the mesoprosopic, leptoprosopic, and hyperleptoprosopic types. A one-way ANOVA was performed to evaluate the mean differences of the maxillary sinus, depending on the FI classification. In this study, the maxillary sinus tended to be wider in those with mesoprosopic type, and tended to be higher in the hyperleptoprosopic type, suggesting a need for clinicians to focus to the shape of the face during clinical treatments.

## 1. Introduction

The maxillary sinus is the largest sinus, and has a pyramidal shape with four walls: the facial, infratemporal, orbital, and nasal surfaces [1]. Its growth is initiated 3 months after birth, and it grows lateral and inferior until the pneumatization of the alveolar bone occurs [2]. Maxillary sinus size is determined around the age of 18 years [3,4].

Since the maxillary sinus is located close to the teeth, research into inflammatory diseases such as dental infections, and also into maxillary sinus floor augmentation during implant surgeries, is being conducted in the dentistry field [5,6,7]. Studies are also being conducted on maxillary sinus diseases in the fields of otolaryngology and plastic surgery, highlighting the importance of the maxillary sinus [5]. The facial skeleton has recently been determined as affecting the maxillary sinus, prompting additional studies on changes in the size of the maxillary sinus [4,8]. However, most studies have been conducted on children, suggesting a need for research on adults.

The facial skeleton is fully developed by adulthood, with the shape varying between individuals and races. To determine this variation, we divided them into euryprosopic type, mesoprosopic type, leptoprosopic type, and hyperleptoprosopic type based on their facial index (FI) classification [9,10]. However, most studies have compared maxillary sinus sizes based on growth, and no research that we know of has analyzed the relationship between maxillary sinus and FI classification. It was therefore necessary to study this undetermined relationship. Most studies have also not measured maxillary sinus size in three dimensions, instead measuring it in 2D, which may be less accurate [2,8,10,11,12,13,14]. The CBCT does not overlap like a panorama [15] and may not produce an accurate representation if pyramid shapes, like the maxillary sinus, are not evaluated in 3D.

This study aimed to determine the size of the maxillary sinus using a 3D program after categorizing South Korean adults according to their FI classification. It was also designed to help prevent clinical surgery complications by observing the relationship between maxillary sinus size and FI classification, including suggesting standard values.

## 2. Materials and Methods

### 2.1. Study Participants

This study analyzed cone-beam computed tomography (CBCT) data of 60 (male 30, female 30) patients aged 20–29 years with malocclusion, no lost teeth, and no asymmetry or systemic diseases who visited the Department of Orthodontics at Dankook University Dental Hospital after being referred from the Department of Oral and Maxillofacial Radiology. The required sample size was calculated using the G*Power 3.1 program (HHU, UK), which determined the sufficiency of the data set.

The radiographic imaging data used in this study were obtained using retrospective analysis. This study was approved by the IRB of Dankook University Dental Hospital (approval no. DUDH IRB 2015-12-022), and the requirement to obtain informed consents from the patients was waived.

### 2.2. Methods

#### 2.2.1. D Image Creation

The CBCT data of the study subjects were collected using a scanner and presented in DICOM format. As the shape and size of the maxillary sinus may differ based on patients’ posture, the participants’ Frankfort horizontal plane (FH) plane was positioned perpendicular to the floor. In addition, the sagittal midline of the face was positioned to align with the imaging device prior to skull radiography. Computed tomography scanning was performed using a 0.39-mm slice increment, 0.39-mm slice thickness, and 512-pixel × 512-pixel matrix (Alphard 3030, Asahi, Kyoto, Japan). For 3D modeling of the maxillary sinus, Mimics (version 22.0, Materialise, Leuven, Belgium) was used to create 3D modeling of DICOM data based on three images: the coronal, sagittal, and frontal views (Figure 1 and Figure 2). To create the space of the maxillary sinus in 3D, the Hounsfield Unit (HU) [16] value, which is a value associated with the Gray Scale, was adjusted. The Hounsfield scale was used to set the standard that falls under the normal range of the bones and soft tissues within the Mimics software. The masking work was performed with values ranging from a minimum of 1024 HU to a maximum of 302 HU. Soft tissues excluding the maxillary sinus were removed using Boolean operation, followed by applying the edit mask function to separate the maxillary sinuses and generate masks for each of the right and left structures (Figure 2). Masking was also implemented with the values of the patient’s skeleton, ranging from a minimum value of 277 HU, to a maximum value of 3071 HU, in relation to the FI classification. The edit mask function was used to remove unnecessary soft tissues, followed by masking. The CalCulate Part function was used to convert the 3-D modelled mask data of the processed maxillary sinus and skull into STL files. Subsequently, the Distance function was used to measure the length of each structure. The masking-work of patient data was then produced in 3D. The produced 3D modeling data were measured item by item after extraction by utilizing the CalCulate Part function to convert it into an STL file.

#### 2.2.2. Measurement Items

The measurement method of this study is thus composed mainly of two parts. The measurements were made using Mimics. The highest point was noted as the standard for measurement. First, measurements were taken to classify the FIs of study participants. The facial height (FH) and facial width (FW) were measured. FH measurement involved measuring the length between the starting point of the nose (N) and the lowest point of the lower jaw (Gn). FW measurement involved measuring the distance between the outermost points of the zygomatic arch (Zy).

Second, the size of the maxillary sinus was measured using measurements of the distance between the orbital cavity (IOF), the distance between the left and right maxillary sinuses (Right-Left sinus distance), and the width and height of the maxillary sinus. All measurements were evaluated by calculating the mean value (reliability is Cronbach’s α = 0.702) following the measurements made by Lee and Park. They were measured according to the FI classification, as follows [10]:N: starting point of the nose.Gn: lowest point of the lower chin border on the midline.Zy: most-lateral point of the zygomatic arch.FH (facial height): distance between N and Gn.FW (facial width): distance between the Zy points.The formula for FI was as follows (Table 1):
(1)$$\mathrm{FI}\;=\frac{\mathrm{FH}}{\mathrm{FW}}\times 100$$

The measured items for the maxillary sinus were as follows (Figure 3 and Figure 4):IOF: distance between the infraorbital foramen from one side to the other.Right–left sinus distance: distance between the left and right maxillary sinuses.Width: corona-view width.Length: sagittal-view widthHeight: maxillary sinus height.Volume: maxillary sinus volume.

#### 2.2.3. Statistics

The measurement items were analyzed using SPSS software (version 23.0, IBM Corporation, Armonk, NY, USA). Since the sample was small, a one-way ANOVA test was conducted to detect significance after testing for normality. Post-hoc analysis was performed on the 95% confidence interval to determine the mean differences between maxillary sinus sizes based on FI classifications. Post hoc analysis was performed using the Scheffe test. Linear regression analysis was also performed to determine the effect of FI classification on maxillary sinus size. A *p*-value of <0.05 was considered significant in all test results.

## 3. Results

Upon categorizing the subjects based on their FI classification, there were four subjects with the normal (i.e., mesoprosopic) facial type. The other 14 and 42 subjects were classified with leptoprosopic and hyperleptoprosopic facial types, respectively. The euryprosopic facial type was not identified in any subject. Table 2 lists the results of comparing maxillary sinus sizes according to FI classifications (Figure 5).

The right–left sinus distance was measured to be Mesoprosopic type 94.79 mm, Leptoprosopic type 89.85 mm, and Hyperleptopros type 88.96 mm, in the order Mesoprosopic type > Leptoprosopic type > Hyperleptopros type. Nevertheless, the differences were not statistically significant. The IOF was measured to be Mesoprosopic type 50.24 mm, Leptoprosopic type 54.47 mm, and Hyperleptopros type 51.17 mm, in the order Leptoprosopic type > Hyperleptopros type > Mesoprosopic type (<0.05). In terms of width, measurements of Mesoprosopic type 30.05 mm, Leptoprosopic type 27.34 mm, and Hyperleptopros type 28.37 mm (Mesoprosopic type > Hyperleptopros type > Leptoprosopic type; <0.05) were observed on the left side, while measurements of Mesoprosopic type 31.05 mm, Leptoprosopic type 29.93 mm, and Hyperleptopros type 27.46 mm (Mesoprosopic type >Leptoprosopic type > Hyperleptopros type; <0.05) were observed on the right side. In terms of length, measurements of Mesoprosopic type 40.94 mm, Leptoprosopic type 38.58 mm, and Hyperleptopros type 40.13 mm (Mesoprosopic type > Hyperleptopros type > Leptoprosopic type) were observed on the left side, while measurements of Mesoprosopic type 41.29 mm, Leptoprosopic type 38.63 mm, and Hyperleptopros type 39.94 mm (Mesoprosopic type > Hyperleptopros type > Leptoprosopic type) were observed on the right side. These measurements, however, were not statistically significant. The height was measured to be Mesoprosopic type 44.74 mm, Leptoprosopic type 43.22 mm, and Hyperleptopros type 49.17 mm (Hyperleptopros type > Mesoprosopic type > Leptoprosopic type; <0.001) on the left side and Mesoprosopic type 46.02 mm, Leptoprosopic type 44.43 mm, Hyperleptopros type 47.49 mm (Hyperleptopros type > Mesoprosopic type > Leptoprosopic type; <0.001) on the right side. The volume of the left frontal sinus for the mesoporsopic, leptoprosopic, and hyperleptoprosopic facial types were 20,004, 19,327, and 22,648 mm^3^, respectively, indicating that the volume was the highest for the hyperleptoprosopic facial type, followed by the mesoprosopic and leptoprosopic facial types (*p* < 0.05). The volume of the right frontal sinus for the mesoprosopic, leptoprosopic, and hyperleptoprosopic facial types were 20,943, 19,096, and 22,429 mm^3^, respectively, indicating that the volume was the highest for the hyperleptoprosopic facial type, followed by the mesoprosopic and leptoprosopic facial types (*p* < 0.05).

Simple linear regression analysis was performed to determine whether the FI classification affected the maxillary sinus size (Table 3). The regression model was considered suitable since F = 4.799 (*p* < 0.05), and the explanatory power was 43% with R^2^ = 0.429. *β* was −0.112 for right–left sinus distance and −0.206 for IOF, and these parameters were not affected by FI classification (*p* > 0.05). On the left, β was −0.189, 0.119, and 0.645 for the width, length, and height, respectively, which were all affected by FI classification (*p* < 0.001); the corresponding values on the right were −0.100, 0.078, and 0.185, and none of these values were affected by the FI classification (*p* > 0.05). *β*(+) was shown in terms of the length on the left and the height on both sides, indicating changes based on the FI classification (Table 3).

## 4. Discussion

The maxillary sinus, which is the largest of the four sinuses, occupies a large region of the face [11], and has therefore been the focus of studies in various clinical fields [1,2,3,4,5,6,7,8,9,10,11,12,13,14]. Until now, most studies have been clinical case studies based on sex, but studies have been conducted recently on the growth of the maxillary sinus [2,8,11,12,13,14]. According to a study by Jun [4], the maxillary sinus grows until adolescence and the twenties in females and males, respectively. Studies of adults are therefore important considering the lack of literature focusing on this population. While most clinical treatments are performed on adult patients, there have yet to be studies comparing size of the maxillary sinus according to the facial skeletons of adults.

The development and anatomy of the facial skeleton depends on several factors, such as sex, race, socioeconomic status, nutrition, and genetics [9]. These factors are essential for planning orthodontic and other various treatments, and are helpful in predicting potential changes [17]. In particular, facial-skeletal measurements can identify racial differences, and are also useful in the anthropology and forensic science fields [18]. Until now, Angle’s classification has been used for facial skeletal measurements; however, since Angle’s classification was designed to analyze malocclusion, it would not be a suitable method for the classification of facial types due to its limitations [9]. A study by Endo [19] indicated that there was no significant difference between sexes in the maxillary sinus size based on Angle’s classification, but there were size differences according to measurements of the tooth face shape. Despite the need to analyze the maxillary sinus size according to the FI classification, no studies have yet been conducted on this topic.

This study classified 60 subjects based on the FI. They comprised 4 mesoprosopic type, 14 leptoprosopi type, 42 hyperleptoprosopic type, and no euryprosopic subjects. The maxillary sinus size was therefore only compared in subjects of mesoprosopic type, leptoprosopic type, and hyperleptoprosopic type.

About 6.7% (6.7% is value rounded of 6.777%) of subjects were of the mesoprosopic type. Upon comparing maxillary sinus size, these subjects had the largest right–left sinus distance, width, and length. In a study by Jahanshahi [20], which also compared skull size according to FI classification, the distances between the cheekbones, nose width, and mouth width were larger in mesoprosopic type than leptoprosopic subjects. Therefore, as for maxillary sinus size, the right–left sinus distance, width, and length values of the mesoprosopic type, the midface size of this type was expected to be the largest. Therefore, among the three facial types, the right–left sinus distance, width, length, and size of the maxillary sinuses are expected to be larger in individuals with the mesoprosopic facial type. Therefore, clinical treatment methods associated with managing the height of the maxillary sinuses would require additional care for individuals with the mesoprosopic facial type.

About 23.3% (23.3% is value rounded of 23.333%) of the 60 subjects were of the leptoprosopic type. Upon comparing maxillary sinus sizes, these subjects had the largest IOF. Kassab [21] reported that the interpupillary distance, canine arc distance, and incisal width of the central incisor were larger in the leptoprosopic type than the mesoprosopic type. Sinavarat [22] reported that there was a strong correlation between the interpupillary distance and canine arc distance. Therefore, considering the maxillary sinus size, the IOF of the leptoprosopic type, which causes a narrow face, was expected to be the largest among the three types. Clinical treatment methods associated with managing the width of the maxillary sinuses would require additional care for individuals with the leptoprosopic facial type.

About 70% of the 60 subjects belonged to the hyperleptoprosopic type. Upon comparing maxillary sinus sizes, these subjects had the largest height. Malim [23] reported that the hyperleptoprosopic type had a larger lower part of the face than the leptoprosopic type. The subject with the longest face in the hyperleptoprosopic type was therefore expected to have the largest facial height among the three types. Moreover, clinical treatment methods associated with managing the width of the maxillary sinuses and orbital cavity would require additional care for individuals with the hyperleptoprosopic facial type.

This study also aimed to determine the association between the facial skeleton and the maxillary sinus. Regression analysis indicated that the FI classification did not affect maxillary sinus size. In addition, the length on the left and the height on both sides were β, indicating that there were differences according to the FI classification. The study by Uchida [24] and Hong [2] indicated that the length and height were correlated with changes in the maxillary sinus volume. Moore [25] similarly reported that changes in the maxillary sinus volume based on age and sex were similar to the changes associated with body growth, such as height and the development of the wrist bones. It therefore seems possible that the maxillary sinus size changes with the size of the facial skeleton.

## 5. Conclusions

This study aims to evaluate the size of the maxillary sinus in Korean adults based on their FI classification, and establish a standard according to the size of the maxillary sinus. Thus, the maxillary sinus and skull of 60 participants (30 women, 30 men) were 3D-modelled using the Mimics program. The data were then converted into an STL file and measurements of the maxillary sinus and skull were taken. The results are as follows.

Based on the file index classification, four subjects were classified with the mesoprosopic facial type and the other 14 and 42 subjects with the leptoprosopic and hyperleptoprosopic facial types, respectively. The euryprosopic facial type was not identified in any subject. The distance between the two maxillary sinuses was determined to be wide for subjects with the mesoprosopic facial type. In the case of the hyperleptoprosopic facial type, the height and volume were found to be large. We propose that facial measurements should be examined before considering any significant surgery involving these areas. In addition, since the FI classification has revealed an association with the maxillary sinus size, this should be studied further. The results of this study should help to prevent complications during various clinical treatments, and provide valuable data for future research on maxillary sinus growth.

## Figures and Tables

**Figure 1 ijerph-19-05040-f001:**
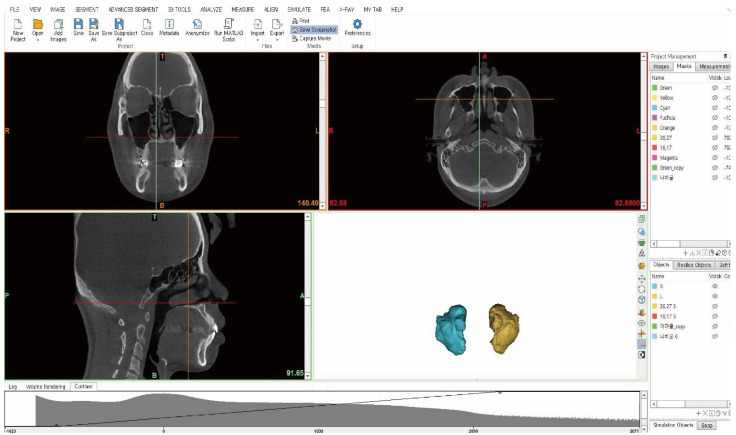
Mimics (version 22.0, Materialise, Leuven, Belgium).

**Figure 2 ijerph-19-05040-f002:**
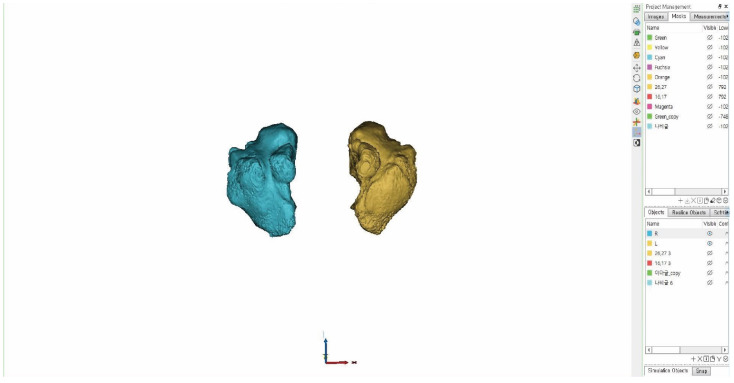
Maxillary sinus 3D.

**Figure 3 ijerph-19-05040-f003:**
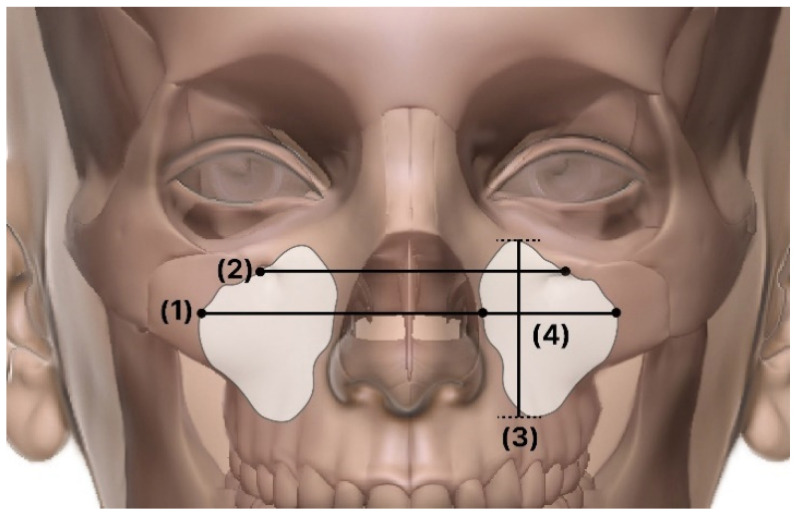
Coronal view. (**1**) IOF: The distance between the infraorbital foramen on one side to the other; (**2**) Right–left sinus distance: The distance between the left and right maxillary sinuses; (**3**) Height: The height of the maxillary sinus; (**4**) Width: The width from the corona view.

**Figure 4 ijerph-19-05040-f004:**
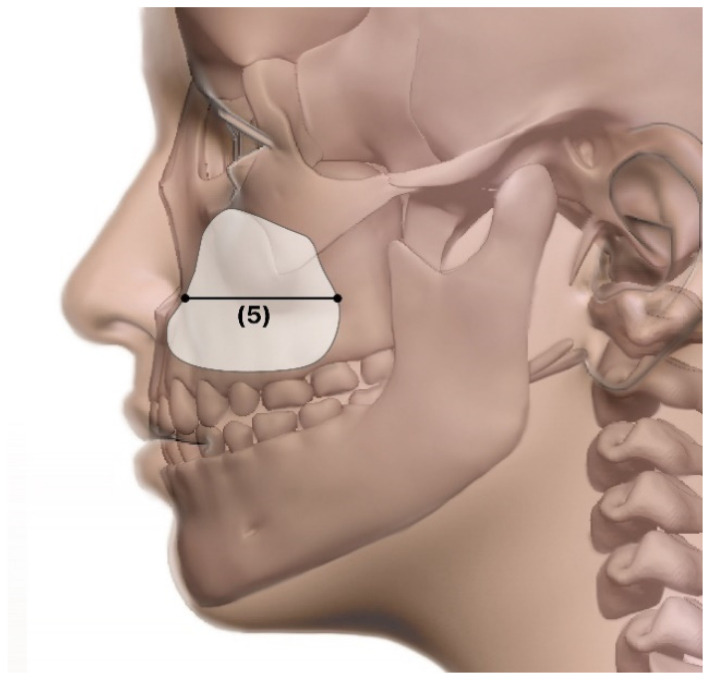
Sagittal view. (**5**) Length: The width from the sagittal view.

**Figure 5 ijerph-19-05040-f005:**
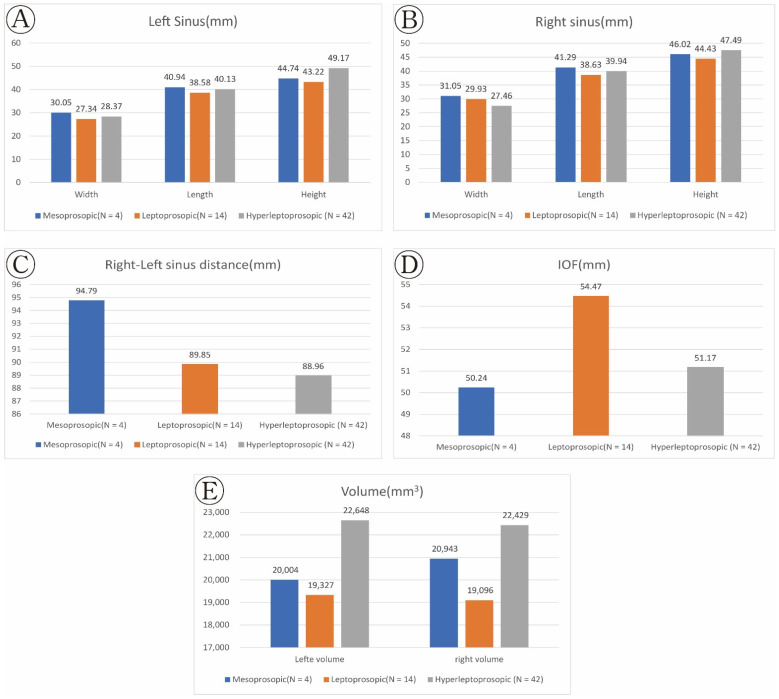
The results of comparing maxillary sinus sizes according to FI classifications. (**A**) Right scheme (**B**) left sinus: width, length, height; (**C**) Right–left sinus distance; (**D**) IOF: The distance between the infraorbital foramen on one side to the other; (**E**) Volume: Right–left sinus volume.

**Table 1 ijerph-19-05040-t001:** Classification lists the number of subjects classified by FI.

Facial Type	Range of FI	N
Mesoprosopic	84.0–87.9	4
Leptoprosopic	88.0–92.9	14
Hyperleptoprosopic	≥93.0	42

**Table 2 ijerph-19-05040-t002:** Comparison of maxillary sinus size according to FI classification.

Measurements		Mesoprosopic (N = 4)	Leptoprosopic (N = 14)	Hyperleptoprosopic (N = 42)	*p*-Value
Right–left sinus distance (mm)		94.79 (3.13)	89.85 (6.66)	88.96 (8.42)	>0.05
IOF (mm)		50.24 (2.83)	54.47 (5.42)	51.17 (4.00)	<0.05 *
(Left)	Width (mm)	30.05 (1.28)	27.34 (1.96)	28.37 (1.87)	<0.05 *
	Length (mm)	40.94 (2.37)	38.58 (3.51)	40.13 (4.00)	>0.05 *
	Height (mm)	44.74 (1.23)	43.22 (2.88)	49.17 (4.05)	<0.001 **
	Volume (mm^3^)	20,004 (2156.57)	19,327 (3813.09)	22,648 (4953.64)	<0.05 *
(Right)	Width (mm)	31.05 (1.91)	29.93 (1.76)	27.46 (4.47)	<0.05 *
	Length (mm)	41.29 (1.10)	38.63 (3.33)	39.94 (4.32)	>0.05
	Height (mm)	46.02 (1.36)	44.43 (1.48)	47.49 (2.66)	<0.001 *
	Volume (mm^3^)	20,943 (1414.81)	19,096 (1219.49)	22,429 (4086.48)	<0.05 *

Data are mean (standard-deviation values), *p*-value were obtained by One-Way ANOVA, * *p* < 0.05, ** *p* < 0.001.

**Table 3 ijerph-19-05040-t003:** Effect of FI classification on maxillary sinus size.

Measurements (mm)		B	SE	*β*	t (*p*)	f (*p*)	R^2^
Constant		3.208	1.600		2.005 *	4.799	0.429
Right-Left sinus distance		−0.015	0.016	−0.112	−0.934		
IOF		−0.016	0.019	−0.206	−0.836		
(Left)	Width	−0.059	0.056	−0.189	−1.054		
	Length	0.019	0.039	0.119	0.484		
	Height	0.088	0.025	0.645	3.506 **		
(Right)	Width	−0.015	0.029	−0.100	−0.522		
	Length	−0.012	0.037	−0.078	−0.321		
	Height	0.007	0.035	0.029	0.185		

*p*-value were obtained by simple linear regression, * *p* < 0.05, ** *p* < 0.001.

## Data Availability

Original data are available upon request to the corresponding author.

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
