# Peer review of "Three-Dimensional CBCT Based Evaluation of the Maxillary Sinus by Facial Index"

_ijerph, 2022, doi:10.3390/ijerph19095040_

Round 1

Reviewer 1 Report

The authors titled the work as a three-dimensional study. However, lengths of the sinus are measured (two-dimensionally) in three planes. 

The measurement of the sinus is not substantiated, some points that do not represent anatomical landmarks (eg RMA-LMA) appear to be random and not reproducible. Why wasn't volume measured as well?
The study population is not heterogeneous and the distribution based on Facial Index appears to be biased. No explanation was given on the normal Facial Index distribution. No statistical corrections for the study population follow.
Acronyms are not used stringently in text or tables.

Figure 1 and 2 are not clear. Table 2 has no unit of measurement and unclear text accompanying it as a footnote.
Sinus anatomy would be expected to match face shape, this study seeks to confirm this metrically. However, no further plausibility is considered in the discussion. The clinical importance of the study is not adequately addressed or exemplified.
Confusing correlation between dental classification of molar relations and facial anatomy.
The percent of the study population given in the discussion does not match the results (page 6 lines 204-211).

Author Response

Thank you so much for commenting on my research.

We have corrected contents according to the reviewer comments.

Please see below File and reply if there is any comments.

Reviewer 2 Report

It is interesting manuscript on the dimensional comparison between maxillary sinus based on the facial index. In this manuscript, the authors used 3D program, the measurement methods and results should be described more precisely.

There are some missing points.

  1. For the better comprehension,  figures or illustrations on the anatomy of the maxillary sinus or Mimics software.
  1. Measurement unit should be added in Table 2 and Table 3 (Example:  RMA-LMA (mm)).
  1. It is necessary to unify the measured items in this study.

Author Response

(The authors gave the same response as above.)

Reviewer 3 Report

The article referenced as ijerph-1657498, entitled “Three-dimensional CBCT based evaluation of the maxillary sinus by facial index” by Jeong-Hyun Lee and Jong-Tae Park used selected DICOM images from CBCT to compare the size and correlation of maxillary sinuses of 60 south Korean male and female adults with their facial index.  This study concluded that maxillary sinus tend to be wider in those with mesoprosopic type, and higher in the hyperleptoprosopic type, suggesting a need for clinicians to focus to the shape of the face during clinical treatments.

Considering the ever-growing interest in the maxillary sinus affecting dental implant, orthodontics and surgical treatment planning, this article is interesting and documented important references for future investigations.

Considering methodology and style of this article, I would like authors respond to the following comments:

  • The statistical analysis using one-way ANOVA is not clearly explained. Did authors actually use Kruskal-Wallis nonparametric test? If the answer is yes, I would remove all references to one-Way ANOVA, because it is misleading and indicates a parametric test. Also, what type of post-hoc test was used?
  • Authors obtained 3D CBCT images, but measurements were done in 2 dimensional DICOM images. Were there any reasons for not using 3D volumetric measurements of sinus instead of 2D linear measurements?
  • The exact reference points for 2D measurements of sinus are vague. Did authors select the widest points in a specific plane?
  • Data presented as tables; however, such data always have much better visual impact if presented as graphs (bar graphs with SD). I would suggest authors to consider presenting their data graphically.
  • How were the Hounsfield Unit ranges selected for masking images? Was in arbitrary?
  • Conclusion section appears to be a summary rather than main conclusions. Please rewrite to reflect only main findings.

Author Response

Thank you so much for commenting on my research.

We have corrected contents according to the reviewer comments.

Please see below File and reply if there is any comments.

This manuscript is a resubmission of an earlier submission. The following is a list of the peer review reports and author responses from that submission.